# Virtual reality for stress management and burnout reduction in nursing: A systematic review protocol

**Ravi Shankar** [1]*, **Fiona Devi Siva Kumar**[1], **Anjali Bundele**[1,2], **Amartya Mukhopadhyay**[1,3]

**1** Research and Innovation, Medical Affairs, Alexandra Hospital, National University Health System, Singapore, **2** Yong Loo Lin School of Medicine, National University of Singapore, Singapore, **3** Division of Respiratory & Critical Care Medicine, Department of Medicine, National University Health System, Singapore

* Ravi_SHANKAR@nuhs.edu.sg

## Abstract

### Background

Burnout is a pervasive issue in the nursing profession, with detrimental consequences for nurses' well-being, patient care, and healthcare systems. Virtual reality (VR) is a promising tool for delivering immersive and engaging interventions to manage stress and reduce burnout. This systematic review aims to synthesize the evidence on the effectiveness of VR interventions for stress and burnout in nursing, characterize the specific intervention approaches, and guide future research and practice.

### Methods

We will search for published and unpublished studies in PubMed, Web of Science, Embase, CINAHL, MEDLINE, The Cochrane Library, PsycINFO, and Scopus from database inception to the present. Randomized controlled trials, quasi-experimental studies, and pre-post studies examining VR interventions for stress and/or burnout in licensed nurses will be included. Two reviewers will independently screen studies, extract data, and assess risk of bias using the Cochrane Risk of Bias 2 tool for randomized trials and the ROBINS-I tool for non-randomized studies. If appropriate, meta-analysis will be performed to estimate pooled effects on stress and burnout outcomes. Subgroup and sensitivity analyses will explore the influence of intervention characteristics and study quality. Narrative synthesis will be conducted if quantitative synthesis is not possible. The review protocol follows the PRISMA-P guidelines and is registered in PROSPERO.

### Discussion

This systematic review will provide a comprehensive synthesis of the evidence on VR interventions for stress and burnout management in nurses. By critically appraising the research and identifying the most promising approaches, the review will guide the development and implementation of evidence-based VR programs to support nurses' well-being and address the urgent problem of burnout. The findings will also identify gaps in

**Data availability statement:** No datasets were generated or analysed during the current study. All relevant data from this study will be made available upon study completion. The completed systematic review data will be deposited in a public repository such as Open Science Framework (OSF) or Figshare, and the DOI will be provided in the final published manuscript.

**Funding:** The author(s) received no specific funding for this work.

**Competing interests:** The authors have declared that no competing interests exist.

the literature and directions for future research to optimize the design and delivery of VR interventions for this high-need population.

**Systematic review registration:** PROSPERO CRD42024604179

## Background

Burnout is a widespread and persistent issue in the nursing profession, affecting nurses across settings and specialties worldwide. Defined as a syndrome of emotional exhaustion, depersonalization, and reduced personal accomplishment resulting from chronic work stress [1], burnout has reached concerning levels among nurses. Based on a comprehensive systematic review and meta-analysis of 85 studies involving 288,581 nurses across 32 countries, the mean prevalence rate of burnout was 30.7% (SD 9.7%) among nurses, with study-specific ascertainments [2]. The mean age of affected nurses was 33.9 years (SD 2.1), with 82.7% being female [2]. Other systematic reviews have estimated nurse burnout prevalence ranging from 11% to 56% [3, 4], with the highest rates observed in critical care, emergency, and oncology settings [5, 6]. Risk factors include younger age [7], male sex [8], single or divorced marital status [8], not having children [8], low staffing levels [9], and workplace violence [10]. The wide prevalence range reflects the diversity of measurement tools, cut-off criteria, and practice environments, but underscores the pervasiveness of the problem.

Burnout takes a heavy toll on nurses' physical, mental, and occupational well-being. Emotionally exhausted and disengaged nurses are at elevated risk for depression, anxiety, sleep disturbances, substance abuse, and suicidal ideation [11,12]. Burnout is associated with poorer physical health, including higher rates of musculoskeletal disorders, respiratory infections, and cardiovascular disease [13]. These health consequences often compel nurses to reduce work hours or leave the profession entirely. Nurse burnout is a strong predictor of job dissatisfaction, absenteeism, and turnover [14], exacerbating staffing shortages and increasing healthcare costs.

Beyond harming nurses themselves, burnout poses a major threat to patient safety and care quality. Robust evidence links nurse burnout to higher rates of medical errors, healthcare-associated infections, and patient mortality [15,16]. Burned-out nurses are less attentive to patients' needs, provide lower-quality teaching and discharge planning, and have poorer communication with patients and colleagues [17]. Patients cared for by burned-out nurses report lower satisfaction, less trust in providers, and more frequent adverse events [18]. As burnout drives experienced nurses out of the bedside, it erodes the overall competency of the nursing workforce.

Healthcare organizations are grappling with the burnout crisis, recognizing its impact on quality of care, patient experience, and financial performance. Burnout-related turnover is estimated to cost the average hospital $4.4 to $6.9 million annually [19]. Initiatives to improve the practice environment, such as adequate staffing, supportive leadership, and meaningful recognition, are essential but often constrained by resource limitations [20]. Individually-focused interventions are also needed to enhance nurses' resilience and coping capacities.

Stress management programs have shown promise in reducing burnout among nurses [21]. Mindfulness-based interventions, cognitive-behavioral training, and relaxation techniques can alleviate emotional exhaustion and increase personal accomplishment [22–24]. However, traditional programs are often time- and location-dependent, posing barriers for participation. The de-stressing effects may also diminish quickly if not reinforced by regular practice. Digital mental health tools, such as web-based and smartphone apps, offer more flexibility but are often limited in engagement and adherence [25].

Virtual reality (VR) has emerged as a novel and immersive platform for delivering mental health interventions. VR refers to computer-generated simulations of three-dimensional environments that can be interacted with in a seemingly real or physical way using specialized equipment like head-mounted displays [26]. By providing multisensory stimuli and rich user engagement, VR can induce potent and lasting therapeutic effects. Accumulating research demonstrates the efficacy of VR in treating anxiety disorders, depression, phobias, and post-traumatic stress [27,28].

Several features of VR make it particularly well-suited for stress management and burnout prevention. First, the immersive and engaging nature of VR can provide a sense of presence and psychological detachment from real-world stressors, allowing for mental restoration [29]. Second, VR environments can be specifically designed to promote relaxation, mindfulness, and self-regulation through calming visual imagery, soothing sounds, and guided meditation [30]. Third, VR simulations of natural environments, such as forests, beaches, and mountains, can provide restorative effects on stress and cognitive fatigue [31]. Fourth, VR interventions can be conveniently delivered in workplace settings and accessed on demand, overcoming participation barriers [32].

A growing body of research has explored the application of VR for stress management in various populations, with promising results. For example, a meta-analysis of 30 studies found that VR-based interventions were effective in reducing stress and anxiety in healthy adults, with a medium effect size [33]. Another systematic review concluded that VR relaxation training was efficacious in reducing stress-related symptoms in young adults [34]. Recent studies have also demonstrated the feasibility and acceptability of VR interventions for occupational stress in healthcare workers [35–38].

Preliminary evidence suggests that VR may be a valuable tool for mitigating stress and burnout among nurses specifically. A study demonstrated that a 10-minute VR relaxation intervention reduced perceived stress by 39.9% among ICU nurses during the COVID-19 pandemic, with 62% of participants reporting the intervention as helpful for stress reduction [39]. Another study showed that a VR mindfulness program resulted in improved relaxation and reduced burnout symptoms in nursing students [40]. However, the literature on VR interventions for nurse burnout is still nascent and scattered. To date, no systematic review has comprehensively synthesized and critically appraised the evidence on this topic.

Given the pressing need for effective and scalable interventions to mitigate the burnout crisis in nursing, it is timely to systematically review the evidence on VR-based approaches. A comprehensive synthesis of the literature will provide valuable insights into the feasibility, acceptability, and efficacy of VR interventions for managing stress and reducing burnout among nurses. By identifying the most promising VR intervention designs and delivery methods, the review will inform the development of evidence-based programs to support nurse well-being. The review will also elucidate gaps and limitations in the current research and guide future studies to optimize the implementation and impact of VR interventions.

## Objectives

The overarching aim of this systematic review is to synthesize and appraise the evidence on VR interventions for stress management and burnout reduction in nursing. The specific objectives are:

1. To identify, describe, and critically evaluate studies that have examined the effects of VR interventions on stress, burnout, and related outcomes among nurses.

2. To characterize the types of VR interventions that have been studied, including the VR content, delivery format, dose, and theoretical basis.

3. To quantify the effects of VR interventions on stress, burnout, and associated outcomes in nurses, and explore potential moderators of effectiveness.

4. To assess the certainty of the evidence and identify sources of bias, inconsistency, and imprecision in the literature.

5. To generate recommendations for future research and practice to optimize the design, implementation, and evaluation of VR interventions for nurse stress and burnout.

By addressing these objectives, the systematic review will provide a rigorous and comprehensive evidence synthesis to guide research, policy, and practice efforts to address the critical issue of burnout in nursing.

## Methods

This systematic review protocol has been developed in accordance with the Preferred Reporting Items for Systematic Review and Meta-Analysis Protocols (PRISMA-P) statement [41] (S1 Checklist). The protocol has been registered in the International Prospective Register of Systematic Reviews (PROSPERO) (registration number: CRD42024604179).

### Eligibility criteria

Studies will be selected based on the following PICOS (Population, Intervention, Comparison, Outcome, Study design) criteria

### Population

The review will include studies of licensed nurses working in any healthcare setting, including hospitals, clinics, long-term care facilities, and community health centers. Nurses may be of any specialty and at any career stage. Studies of nursing students, nursing assistants, or other healthcare professionals will be excluded.

### Intervention

The review will include studies that evaluate any VR-based intervention designed to manage stress, reduce burnout, or improve coping and resilience among nurses. VR interventions are defined as immersive, interactive simulations of real or imagined environments delivered via a head-mounted display or other specialized equipment. Eligible VR interventions may incorporate various approaches, such as:

- Relaxation and meditation exercises

- Guided imagery and visualization

- Simulations of natural environments

- Biofeedback and self-regulation training

- Cognitive-behavioral stress management techniques

- Mindfulness-based practices

VR interventions may be delivered in a single session or over multiple sessions, and may be provided in individual or group formats. Studies that use non-immersive digital interventions (e.g., web-based programs, smartphone apps) or only measure VR outcomes in a non-intervention context will be excluded.

## Comparison

Studies may compare the VR intervention to no intervention, waitlist control, treatment as usual, or an alternative active intervention (e.g., in-person stress management program, web-based education). Studies without a comparison group will also be included.

## Outcomes

The primary outcomes of interest are stress and burnout. Stress may be measured using self-report questionnaires (e.g., Perceived Stress Scale [42], Nursing Stress Scale [43]), physiological indicators (e.g., cortisol, heart rate variability), or behavioral observations (e.g., coping strategies). Burnout may be assessed using validated instruments such as the Maslach Burnout Inventory [1], Oldenburg Burnout Inventory [44], or Copenhagen Burnout Inventory [45]. Secondary outcomes may include psychological well-being, resilience, job satisfaction, and turnover intention.

For the Perceived Stress Scale, scores ≥ 26 will be considered high stress. For the Maslach Burnout Inventory, we will use established cut-offs: emotional exhaustion ≥ 27, depersonalization ≥ 10, and personal accomplishment ≤ 33. For other validated instruments, we will use published cut-off scores when available or author-defined thresholds with clear justification.

## Study designs

The review will include randomized controlled trials (RCTs), quasi-experimental studies, and single-group pre-post studies. Observational studies (e.g., cohort, case-control) will be excluded. Systematic reviews, meta-analyses, qualitative studies, and non-empirical articles (e.g., commentaries, editorials) will be excluded, but their references will be screened for eligible primary studies. Qualitative studies are excluded as our primary aim is to quantitatively assess the effectiveness of VR interventions. While we acknowledge that qualitative insights are valuable, including them would require different methodological approaches beyond the scope of this review.

## Search strategy

A comprehensive search strategy will be developed in consultation with a health sciences librarian to ensure thoroughness and accuracy. The search will cover a wide range of electronic databases, including PubMed, Web of Science, Embase, CINAHL, MEDLINE, The Cochrane Library, PsycINFO, and Scopus, from their inception to the present. This approach aims to capture all relevant literature and ensure the inclusivity of the review process.

The search strategy will use combinations of keywords and controlled vocabulary terms (e.g., MeSH) related to the key concepts of virtual reality, nurses, stress, and burnout. An example search string for PubMed is:

("virtual reality" OR VR OR "head-mounted display" OR HMD OR immersive OR "computer simulation") AND (nurs* OR "nursing staff" OR "nursing personnel") AND (stress OR burnout OR "occupational stress" OR "compassion fatigue" OR "mental health" OR resilience OR coping).

The search strategy will be adapted to the syntax and indexing of each database. No language or publication date filters will be applied. The full search strings for each database will be provided in the review appendix.

To identify grey literature, we will search Google Scholar, ProQuest Dissertations & Theses, and conference proceedings from relevant nursing and healthcare organizations (e.g., Sigma Theta Tau International, American Nurses Association). We will also hand-search the

references of included studies and relevant systematic reviews. Finally, we will contact experts in the field to inquire about ongoing or unpublished studies.

### Study selection

The results of the database searches will be exported into Covidence systematic review software [46] for deduplication and screening. Two reviewers will independently screen the titles and abstracts of all records against the eligibility criteria. Studies that clearly do not meet the criteria will be excluded at this stage. The full texts of the remaining articles will be retrieved and independently assessed by two reviewers. Any disagreements on study inclusion will be resolved through discussion or consultation with a third reviewer. Reasons for excluding full-text articles will be recorded and reported in a PRISMA flow diagram [47].

## Data extraction

Data extraction will be conducted using a standardized form in Covidence software, following a pilot testing phase to refine the extraction process. Two independent reviewers will systematically extract data, with any disagreements resolved through consensus or consultation with a third reviewer. The extraction protocol will capture comprehensive information across five key domains: study characteristics (including authorship, publication details, funding, and conflicts of interest), participant demographics (encompassing sample size, age, gender distribution, ethnicity, nursing specialization, work setting, and professional experience), intervention specifics (detailing VR implementation, delivery methods, dosage, duration, theoretical framework, and control conditions), outcome measurements (including assessment tools, timing, statistical results, and adverse events), and methodological considerations (covering randomization procedures, blinding methods, participant retention, data handling, and statistical power) (S1 File). If data are unclear or missing, we will contact the study authors for clarification or additional information. If studies report results for multiple follow-up time points, we will extract data for the longest duration of follow-up. If studies include both nurses and other healthcare professionals, we will only extract data for the nurse subgroup. If subgroup data are not reported, we will contact authors to request this information.

### Risk of bias assessment

Two reviewers will independently assess the risk of bias in each study, with disagreements resolved through discussion or consultation with a third reviewer. Randomized controlled trials will be appraised using the Cochrane Risk of Bias 2 (RoB 2) tool [48], which evaluates bias in the randomization process, deviations from intended interventions, missing outcome data, outcome measurement, and selection of reported results. Quasi-experimental and pre-post studies will be assessed using the ROBINS-I tool [49], which covers confounding, selection of participants, classification of interventions, deviations from interventions, missing data, outcome measurement, and selection of reported results.

For each domain of bias, studies will be judged as having low risk, moderate risk, serious risk, or critical risk of bias. An overall risk of bias judgment will be assigned to each study based on the highest risk level across domains. The risk of bias assessments will be used to interpret the results and grade the certainty of the evidence.

For any disagreements in screening, data extraction, or quality assessment, reviewers will first attempt resolution through discussion, referring to the protocol criteria. If consensus cannot be reached, a third reviewer will review the evidence and make a final decision. All disagreements and their resolution will be documented in a decision log.

## Data synthesis

### Narrative synthesis

We will provide a narrative synthesis of the characteristics and findings of included studies, following established guidelines [50]. Studies will be grouped by intervention type, comparison group, and outcome. We will qualitatively summarize the key features of each study, including the participant population, VR intervention content and delivery, and main results. We will also identify patterns, similarities, and differences across studies and explore potential explanations for inconsistencies.

### Quantitative synthesis

If there are at least two studies with comparable interventions, outcomes, and study designs, we will conduct meta-analysis to quantitatively synthesize the results. For continuous outcomes (e.g., stress scores), we will calculate standardized mean differences (Hedges' g) between the VR intervention and comparison groups. For dichotomous outcomes (e.g., presence of burnout), we will calculate risk ratios. All effect sizes will be reported with 95% confidence intervals.

We will begin by visually inspecting forest plots to qualitatively assess heterogeneity patterns before proceeding with statistical testing. We will assess statistical heterogeneity using the $I2$ statistic, which describes the percentage of variation across studies that is due to heterogeneity rather than chance [51]. If $I2$ is greater than 50%, indicating substantial heterogeneity, we will use a random-effects meta-analysis model. If $I2$ is less than 50%, we will use a fixed-effects model. We will also assess clinical and methodological heterogeneity by examining differences in participant characteristics, interventions, outcomes, and study designs.

If there is significant statistical or clinical heterogeneity, we will explore potential sources through subgroup and meta-regression analyses. For meta-regression, we will use restricted maximum likelihood estimation for random effects models, with a minimum requirement of 10 studies to ensure stable estimates. Subgroup analyses will be conducted based on:

- Type of VR intervention (e.g., relaxation, mindfulness, cognitive-behavioral)

- Dose and duration of VR intervention

- Nursing specialty (e.g., critical care, emergency, oncology)

- Healthcare setting (e.g., hospital, long-term care)

- Study design (e.g., RCT, quasi-experimental, pre-post)

- Risk of bias rating (e.g., low, moderate, serious)

Meta-regression will be used to investigate the influence of continuous moderator variables, such as participant age and baseline stress levels. For studies reporting multiple outcome time points, we will conduct sensitivity analyses comparing results using only the longest follow-up versus separate analyses for different time points.

We will assess publication bias through visual inspection of funnel plots and statistical tests such as Egger's regression [52]. If publication bias is suspected, we will use the trim-and-fill method to adjust the effect size estimates [53].

We will conduct leave-one-out analyses to identify potentially influential studies and perform sensitivity analyses excluding these studies to assess their impact on pooled estimates.

To ensure transparency and reproducibility, we will provide detailed documentation of all analytical decisions, including any post-hoc analyses conducted in response to unexpected patterns in the data. The complete R code for all analyses will be made available as a S1 File.

All meta-analyses will be conducted using the Cochrane Review Manager software [54]. If quantitative synthesis is not possible due to insufficient data or heterogeneity, we will present the findings in narrative form.

## Certainty assessment

We will grade the certainty of the evidence for each outcome using the Grading of Recommendations, Assessment, Development and Evaluations (GRADE) approach [55]. The GRADE system assesses the certainty of the body of evidence based on five domains: risk of bias, imprecision, inconsistency, indirectness, and publication bias. The certainty of evidence will be rated as high, moderate, low, or very low. Two reviewers will independently assess each domain and assign an overall GRADE rating, with discrepancies resolved through discussion or input from a third reviewer.

## Ethics and dissemination

Ethical approval is not required for this systematic review, as it will only include published and publicly accessible data. The review findings will be submitted for publication in a peer-reviewed journal and presented at relevant conferences. We will also develop an evidence brief and infographic to disseminate the results to healthcare organizations, nursing associations, and other stakeholders. The datasets generated during the review will be made available upon reasonable request.

## Discussion

This systematic review will comprehensively synthesize and critically appraise the evidence on VR interventions for stress management and burnout reduction in nurses. To our knowledge, this will be the first systematic review to focus specifically on VR-based approaches for this population and purpose. Previous reviews have examined the effectiveness of VR interventions for stress and anxiety in general populations [33,56] or healthcare workers broadly [57], but have not targeted nurses or assessed burnout outcomes. Given the unique stressors and challenges faced by nurses, and the urgent need to mitigate the growing burnout crisis, a focused review is warranted.

The review will have several strengths. First, the broad eligibility criteria will allow for a comprehensive mapping of the VR intervention landscape, including diverse VR modalities, therapeutic approaches, and delivery formats. Second, the inclusion of both published and grey literature will minimize the risk of publication bias. Third, the use of validated tools to assess risk of bias and grade the certainty of the evidence will provide a rigorous and transparent appraisal of the literature. Finally, the involvement of multiple reviewers in screening, data extraction, and quality assessment will enhance the reliability of the findings.

However, we anticipate some limitations. First, the inconsistency in how stress and burnout are defined and measured across studies may hinder comparisons and synthesis. We will use validated instruments as a reference standard and carefully examine the conceptual and psychometric properties of outcome measures. Second, the heterogeneity in VR intervention types and study designs may limit the feasibility and interpretability of meta-analysis. We will follow best-practice guidelines for conducting narrative synthesis and exploring sources of heterogeneity [50]. Third, the generalizability of the findings may be restricted by the inclusion of only English-language studies. However, we do not expect language bias to substantially alter the conclusions, as most VR research is published in English.

The review findings will have important implications for nursing practice, education, and research. By identifying the most promising VR intervention approaches and delivery formats,

the review will inform the development and implementation of evidence-based programs to support nurse well-being. The integration of effective VR tools into nursing education and professional development may help build resilience and coping skills. VR stress management may be particularly valuable for high-risk nursing specialties such as critical care, emergency, and oncology. The portability and flexibility of VR may also extend access to nurses in rural and underserved areas.

The review will also guide future research on VR interventions for nurse stress and burnout. By identifying gaps and limitations in the current evidence base, the review will highlight priority areas for investigation. For example, research is needed to optimize the design features and dose-response of VR interventions, evaluate implementation barriers and facilitators, and assess the long-term sustainability of effects. Economic evaluations are also needed to establish the cost-effectiveness of VR programs compared to traditional stress management approaches.

## Supporting information

**S1 Checklist. PRISMA-P Checklist. Checklist of items to include when reporting a systematic review protocol.**
(DOC)

**S1 File. Supplementary file contains three components: (1) Data extraction form for collecting information from included studies, (2) Risk of bias assessment tools including Cochrane Risk of Bias 2 (RoB 2) tool for randomized trials and Risk of Bias in Non-randomized Studies of Interventions (ROBINS-I) tool, and (3) GRADE evidence profile template and summary tables for assessing certainty of evidence.**
(DOC)

## Author contributions

**Conceptualization:** Ravi Shankar.

**Data curation:** Ravi Shankar, Fiona Devi Siva Kumar.

**Formal analysis:** Ravi Shankar, Fiona Devi Siva Kumar, Anjali Bundele.

**Investigation:** Ravi Shankar.

**Methodology:** Ravi Shankar, Fiona Devi Siva Kumar, Anjali Bundele.

**Project administration:** Ravi Shankar.

**Resources:** Ravi Shankar.

**Software:** Ravi Shankar.

**Supervision:** Ravi Shankar, Amartya Mukhopadhyay.

**Validation:** Ravi Shankar, Anjali Bundele.

**Writing – original draft:** Ravi Shankar.

**Writing – review & editing:** Ravi Shankar, Fiona Devi Siva Kumar, Anjali Bundele, Amartya Mukhopadhyay.

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
