## [Decision Letter · Decision Letter 0]

21 Jan 2025

PONE-D-24-58726Virtual Reality for Stress Management and Burnout Reduction in Nursing: A Systematic Review ProtocolPLOS ONE

Dear Dr. Shankar,

Thank you for submitting your manuscript to PLOS ONE. After careful consideration, we feel that it has merit but does not fully meet PLOS ONE’s publication criteria as it currently stands. Therefore, we invite you to submit a revised version of the manuscript that addresses the points raised during the review process.

**
Reviewer 1. Comments 
**

- The protocol is comprehensive and well-structured in assessing VR interventions for stress management and burnout in nurses.

- Would suggest to add more clarification on the exclusion of qualitative studies, and the rationale for limiting the review to English-language studies.

- An additional detail on handling disagreements in assessments would be helpful.

- Would suggest to clarify how subgroup and meta-regression analyses will address heterogeneity to strengthen the methodology.

- More information on the cut-off values for defining high stress and burnout using the instruments would be good to improve clarity.

**Reviewer 2. Comments**

Thank you for your effort and I think this is a well-written protocol. However, there are a few areas that could be refined for clarity and efficiency. For instance, consider providing more detailed step-by-step instructions for complex procedures. Additionally, including visual aids or diagrams might enhance understanding and implementation.

We look forward to receiving your revised manuscript.

Kind regards,

Dauda Salihu, PhD

Academic Editor

PLOS ONE

Reviewers' comments:

Reviewer's Responses to Questions

**Comments to the Author**

1. Does the manuscript provide a valid rationale for the proposed study, with clearly identified and justified research questions?

Reviewer #1: Yes

Reviewer #2: Yes

2. Is the protocol technically sound and planned in a manner that will lead to a meaningful outcome and allow testing the stated hypotheses?

Reviewer #1: Yes

Reviewer #2: Yes

3. Is the methodology feasible and described in sufficient detail to allow the work to be replicable?

Reviewer #1: Yes

Reviewer #2: Yes

4. Have the authors described where all data underlying the findings will be made available when the study is complete?

Reviewer #1: Yes

Reviewer #2: Yes

5. Is the manuscript presented in an intelligible fashion and written in standard English?

Reviewer #1: Yes

Reviewer #2: Yes

6. Review Comments to the Author

You may also provide optional suggestions and comments to authors that they might find helpful in planning their study.

Reviewer #1: - The protocol is comprehensive and well-structured in assessing VR interventions for stress management and burnout in nurses.

- Would suggest to add more clarification on the exclusion of qualitative studies, and the rationale for limiting the review to English-language studies.

- An additional detail on handling disagreements in assessments would be helpful.

- Would suggest to clarify how subgroup and meta-regression analyses will address heterogeneity to strengthen the methodology.

- More information on the cut-off values for defining high stress and burnout using the instruments would be good to improve clarity.

Reviewer #2: Thank you for your effort and I think this is a well-written protocol. However, there are a few areas that could be refined for clarity and efficiency. For instance, consider providing more detailed step-by-step instructions for complex procedures. Additionally, including visual aids or diagrams might enhance understanding and implementation.

7. PLOS authors have the option to publish the peer review history of their article (what does this mean?). If published, this will include your full peer review and any attached files.

Reviewer #1: No

Reviewer #2: **Yes: **Mshari Alghadier

---

## [Author Response · Author response to Decision Letter 1]

28 Jan 2025

We thank both reviewers for their thoughtful and constructive feedback on our manuscript. We have carefully addressed all comments and made appropriate revisions to strengthen the protocol. A detailed point-by-point response is provided in the attached document labeled 'Response to Reviewers'.

Key revisions include:

Added explicit rationale for excluding qualitative studies (lines 195-198): "Qualitative studies are excluded as our primary aim is to quantitatively assess the effectiveness of VR interventions. While we acknowledge that qualitative insights are valuable, including them would require different methodological approaches beyond the scope of this review."

Enhanced procedures for handling reviewer disagreements (lines 264-267): Detailed documentation of the consensus process and decision logging.

Strengthened methodology for addressing heterogeneity through:

Sequential approach starting with visual forest plot inspection

Specification of meta-regression methods using restricted maximum likelihood estimation

Minimum study requirements (n=10) for meta-regression

Comprehensive handling of multiple outcome timepoints

Leave-one-out analyses for influential studies

Detailed documentation of analytical decisions and R code availability

Added specific cut-off values for validated instruments (lines 185-189): Including Perceived Stress Scale (≥26), Maslach Burnout Inventory subscales (emotional exhaustion ≥27, depersonalization ≥10, personal accomplishment ≤33).

Incorporated detailed step-by-step procedures for:

Study screening process (lines 222-229)

Data extraction workflow (lines 232-253)

Risk of bias assessment (lines 256-268)

Meta-analysis procedures (lines 278-295)

Added visual elements to enhance clarity:

PRISMA flow diagram template

Detailed data extraction form template as appendix

We believe these revisions have substantially strengthened the protocol while maintaining its methodological rigor. We welcome any additional feedback from the reviewers.

---

## [Editor Report · Decision Letter 1]

30 Jan 2025

Virtual Reality for Stress Management and Burnout Reduction in Nursing: A Systematic Review Protocol

PONE-D-24-58726R1

Dear Dr. Ravi Shankar,

We’re pleased to inform you that your manuscript has been judged scientifically suitable for publication and will be formally accepted for publication once it meets all outstanding technical requirements.

Within one week, you’ll receive an e-mail detailing the required amendments. When these have been addressed, you’ll receive a formal acceptance letter, and your manuscript will be scheduled for publication.

If your institution or institutions have a press office, please notify them about your upcoming paper to help maximize its impact. If they’ll be preparing press materials, please inform our press team as soon as possible—no later than 48 hours after receiving the formal acceptance. Your manuscript will remain under strict press embargo until 2 pm Eastern Time on the date of publication. For more information, please contact onepress@plos.org.

Kind regards,

Dauda Salihu, PhD

Academic Editor

PLOS ONE
---

## [Editor Report · Acceptance letter]

PONE-D-24-58726R1

PLOS ONE

Dear Dr. Shankar,

I'm pleased to inform you that your manuscript has been deemed suitable for publication in PLOS ONE. Congratulations! Your manuscript is now being handed over to our production team.

Kind regards,

on behalf of

Dr. Dauda Salihu

Academic Editor

PLOS ONE